# Unsupervised Meta-Learning for Reinforcement Learning

## Abstract

Meta-learning algorithms use past experience to learn to quickly solve new tasks. In the context of reinforcement learning, meta-learning algorithms acquire reinforcement learning procedures to solve new problems more efficiently by utilizing experience from prior tasks. The performance of meta-learning algorithms depends on the tasks available for meta-training: in the same way that supervised learning generalizes best to test points drawn from the same distribution as the training points, meta-learning methods generalize best to tasks from the same distribution as the meta-training tasks. In effect, meta-reinforcement learning offloads the design burden from algorithm design to task design. If we can automate the process of task design as well, we can devise a meta-learning algorithm that is truly automated. In this work, we take a step in this direction, proposing a family of unsupervised meta-learning algorithms for reinforcement learning. We motivate and describe a general recipe for unsupervised meta-reinforcement learning, and present an instantiation of this approach. Our conceptual and theoretical contributions consist of formulating the unsupervised meta-reinforcement learning problem and describing how task proposals based on mutual information can be used to train optimal meta-learners. Our experimental results indicate that unsupervised meta-reinforcement learning effectively acquires accelerated reinforcement learning procedures without the need for manual task design and these procedures exceed the performance of learning from scratch.

## 1 Introduction

Reusing past experience for faster learning of new tasks is a key challenge for machine learning. Meta-learning methods achieve this by using past experience to explicitly optimize for rapid adaptation (Mishra et al., 2017; Snell et al., 2017; Schmidhuber, 1987; Finn et al., 2017a; Gupta et al., 2018; Wang et al., 2016; Al-Shedivat et al., 2017). In the context of reinforcement learning (RL), meta-reinforcement learning (meta-RL) algorithms can learn to solve new RL tasks more quickly through experience on past tasks (Duan et al., 2016b; Gupta et al., 2018; Finn et al., 2017a). Typical meta-RL algorithms assume the ability to sample from a pre-specified task distribution, and these algorithms learn to solve new tasks *drawn from this distribution* very quickly. However, specifying a task distribution is tedious and requires a significant amount of supervision (Finn et al., 2017b; Duan et al., 2016b) that may be difficult to provide for large, real-world problem settings. The performance of meta-learning algorithms critically depends on the meta-training task distribution, and meta-learning algorithms generalize best to new tasks which are drawn from the same distribution as the meta-training tasks (Finn & Levine, 2018). In effect, meta-RL offloads much of the design burden from algorithm design to designing a sufficiently broad and relevant distribution of meta-training tasks. While this offloading helps in acquiring representations for fast adaptation to the specified task distribution, specifying this is often tedious and challenging. A natural question is whether we can do away with manual task design and develop meta-RL algorithms that learn only from unsupervised environment interaction. In this paper, we take an initial step toward the formalization and design of such methods.

Our goal is to automate the meta-training process by removing the need for hand-designed meta-training tasks. To that end, we introduce unsupervised meta-RL: meta-learning from a task distribution that is acquired automatically, rather than requiring manual design of the meta-training tasks. Unsupervised meta-RL methods must solve two difficult problems together: meta-RL with broad task

distributions, and unsupervised exploration for proposing a wide variety of tasks for meta-learning. Since the assumptions of our method differ fundamentally from prior meta-RL methods (we do not assume access to hand-specified meta-training tasks that use human-specified reward functions), the best points of comparison for our approach are learning meta-test tasks entirely from scratch with conventional RL algorithms. Our method can also be thought of as automatically acquiring an *environment-specific* learning procedure for deep neural network policies, somewhat related to data-driven initialization procedures explored in supervised learning (Krähenbühl et al., 2015; Hsu et al., 2018).

The primary contributions of our work are to propose a framework for unsupervised meta-RL; to sketch out a family of unsupervised meta-RL algorithms; to provide a theoretical derivation that allows us to reason about the optimality of unsupervised meta-RL methods in terms of mutual information objectives; and to describe an instantiation of an algorithm from this family that builds on a recently proposed procedure for unsupervised exploration (Eysenbach et al., 2018) and model-agnostic meta-learning (MAML) (Finn et al., 2017a). In addition to our theoretical derivations, we provide an empirical evaluation that studies the performance of two variants of our approach on simulated control tasks. Our experimental evaluation shows that, for a variety of tasks, unsupervised meta-RL can effectively acquire RL procedures that perform significantly better than standard RL methods that learn from scratch, without requiring additional task knowledge.

## 2 RELATED WORK

Our work lies at the intersection of meta-RL, goal generation, and unsupervised exploration. Meta-learning algorithms use data from multiple tasks to learn how to learn, acquiring rapid adaptation procedures from experience (Schmidhuber, 1987; Naik & Mammone, 1992; Thrun & Pratt, 1998; Bengio et al., 1992; Hochreiter et al., 2001; Santoro et al., 2016; Andrychowicz et al., 2016; Li & Malik, 2017; Ravi & Larochelle, 2017; Finn et al., 2017a; Munkhdalai & Yu, 2017; Snell et al., 2017). These approaches have been extended into the setting of RL (Duan et al., 2016b; Wang et al., 2016; Finn et al., 2017a; Sung et al., 2017; Mishra et al., 2017; Gupta et al., 2018; Mendonca et al., 2019; Houthooft et al., 2018; Stadie et al., 2018; Rakelly et al., 2019; Nagabandi et al., 2018a; Houthooft et al., 2018). In practice, the performance of meta-learning algorithms depends on the user-specified meta-training task distribution. We aim to lift this limitation and provide a general recipe for avoiding manual task engineering for meta-RL. To that end, we make use of unsupervised task proposals. These proposals can be obtained in a variety of ways, including adversarial goal generation (Sukhbaatar et al., 2017; Held et al., 2017), information-theoretic methods (Gregor et al., 2016; Eysenbach et al., 2018; Co-Reyes et al., 2018; Achiam et al., 2018), and even random functions. We argue that, theoretically, methods based on mutual information have the potential to provide *optimal* task proposals for unsupervised meta-RL.

Exploration methods that seek out novel states are also closely related to goal generation methods (Pathak et al., 2017; Schmidhuber, 2009; Bellemare et al., 2016; Osband et al., 2016; Stadie et al., 2015), but do not by themselves aim to generate new tasks or learn to adapt more quickly to new tasks, only to achieve wide coverage of the state space. These methods are complementary to our approach, but address a distinct problem. While model-based RL methods (Chua et al., 2018; Srinivas et al., 2018; Deisenroth & Rasmussen, 2011; Nagabandi et al., 2018b; Atkeson & Santamaria, 1997) might likewise use unsupervised experience to learn a dynamics model, they are not equipped with a strategy to employ that model to learn effectively at test time.

Related to our work are prior methods that study the training of goal-conditioned policies (Schaul et al., 2015; Pong et al., 2018; Andrychowicz et al., 2017). Indeed, our theoretical derivation first studies the goal reaching case, before generalizing it to the more general case of arbitrary tasks. This generalization allows unsupervised meta-learning methods to solve arbitrary tasks at meta-test time without being restricted to a task parameterization or goal specification. Additionally, in Section 3.4 we discuss why the framework of meta-learning gives us theoretical benefits over the goal reaching paradigm.

Figure 1: **Unsupervised meta-reinforcement learning**: Given an environment, unsupervised meta-RL produces an environment-specific learning algorithm that quickly acquire new policies that maximize any task reward function.

# 3 UNSUPERVISED META-REINFORCEMENT LEARNING

In this paper, we consider the problem of automatically tailoring (i.e., learning) a reinforcement learning algorithm to solve tasks on a specific environment. This learning algorithm should be meta-learned without requiring any human supervision or specification of task distribution. The implicit assumption in this formulation will be that all test-time tasks share the same dynamics but use different reward functions.

To talk about environments on which many tasks might be performed, we consider a controlled Markov process (CMP) – a Markov decision process without a reward function, $C = (S, A, P, \gamma, \rho)$, with state space $S$, action space $A$, transition dynamics $P$, discount factor $\gamma$ and initial state distribution $\rho$. The CMP along with this reward function $r_z$ produces a Markov decision processes $M_i = (S, A, P, \gamma, \rho, r_z)$. The goal of the learning algorithm $f$ is to learn an optimal policy $\pi_i^*(a \mid s)$ for any reward function $r_z$ that is provided with the CMP. With this notation in place, we formally define the unsupervised meta-RL as follows:

**Definition 1.** *Unsupervised Meta-Reinforcement Learning*: Given a CMP $C$, output a learning algorithm $f$ that can learn new tasks efficiently in MDPs defined by CMP $C$ together with an (unknown) reward function $r_z$.

The data available to unsupervised meta-RL are unsupervised interactions with the CMP, and does not include observing any of the reward functions $r_z$. We illustrate this problem setting in Figure 1. In contrast, in standard meta-reinforcement learning, a task distribution $P(T)$ is known and we simply try to optimize for a fast learning algorithm on that known task distribution. A detailed definition can be found in Section 3.2.

In the rest of this section, we will first sketch out a general recipe for an unsupervised meta-RL algorithm, present a theoretical derivation for an *optimal* unsupervised meta-learning method, and then instantiate a practical approximation to this theoretically-motivated approach using components from recently proposed exploration and meta-learning algorithms.

## 3.1 A GENERAL RECIPE

Our framework, unsupervised meta-RL, consists of a task proposal mechanism and a meta-learning method. Formally, we will define the task distribution as a mapping from a latent variable $z \sim p(z)$ to a reward function $r_z(s, a) : S \times A \to \mathbb{R}^1$. That is, for each value of the random variable $z$, we have a different reward function $r_z(s, a)$. Under this formulation, learning a task distribution amounts to optimizing a parametric form for the reward function $r_z(s, a)$ that maps each $z \sim p(z)$ to a different reward function. The choice of this parametric form represents an important design decision for an unsupervised meta-learning method, and the resulting set of tasks is often referred to as a task or goal proposal procedure. The second component is the meta-learning[2] algorithm, which takes the family of reward functions induced by $p(z)$ and $r_z(s, a)$ along with the associated CMP, and meta-learns a RL algorithm $f$ that can quickly adapt to any task from the task distribution defined by

---

[1]In most cases $p(z)$ is chosen to be a uniform categorical so it is not challenging to specify

[2]A meta-reinforcement learning algorithm learns how to learn: it uses a set of meta-training tasks to learn a learning function f, which can then learn a new task. We refer to this learned learning function f as an "acquired reinforcement learning procedure," following prior work, such as MAML (Finn et al., 2017a) and RL2 (Duan et al., 2016b)

$p(z)$ and $r_z(s, a)$ in the given CMP. The meta-learned algorithm $f$ can then learn new tasks quickly at meta-test time, when a user-specified reward function is actually provided. This generic design for an unsupervised meta-RL algorithm is summarized in Figure 1.

The "no free lunch theorem" (Wolpert et al., 1995; Whitley & Watson, 2005) might lead us to expect that a truly generic approach to proposing a task distribution would not yield a learning procedure $f$ that is effective on any real tasks, or even on the meta-training tasks. However it is important to note that even without a reward function, an unsupervised meta-learning algorithm *can* collect and organize meaningful information about the environment. For example, this information could include policies for reaching certain states, estimates for the distances between states, a map of the environment, or a model of the dynamics. Importantly, the meta-learner is not restricted to learning information that is human-interpretable, especially since human-interpretable representations may not be optimal for fast adaptation. In the following sections, we will discuss how to formulate an optimal unsupervised meta-learner that minimizes regret on new meta-test tasks in the absence of any prior knowledge. We first discuss the notion of an optimal meta-learner and then show how we can train one without requiring task distributions to be hand-specified.

A more general version of this algorithm might also use $f$ to inform the acquisition of tasks, allowing for an alternating optimization procedure the iterates between learning $r_z(s, a)$ and updating $f$, for example by designing tasks that are difficult for the current algorithm $f$ to handle. However, in this paper we will consider the stagewise approach, which acquires a task distribution once and meta-trains on it, leaving the iterative variant for future work.

## 3.2 Optimal Meta-Learners

We first define an abstract notion of an optimal meta-learner, when *given a task distribution*. This definition will be useful to showing that for certain test-time task distributions, meta-training on self-proposed tasks (i.e., unsupervised meta-training) will yield the optimal meta-learner at test-time. We assume that an optimal meta-learner takes in a distribution over tasks for a given CMP and outputs a learning procedure $f$ that minimizes expected regret when learning tasks drawn from the same distribution as seen in meta-training. As before, the task distribution is defined by a variable $z \sim p(z)$ and a reward function $r_z(s, a) : \mathcal{S} \times \mathcal{A} \to \mathbb{R}$. An optimal meta-learner optimizes

$$\min_f \text{REGRET}(f, p(r_z)) = \min_f \mathbb{E}_{z \sim p(z)} \left[ R(f^*, r_z) - R(f, r_z) \right], \qquad (1)$$

where $p(r_z)$ is a distribution over reward functions, parameterized by $z$, $R(f, r_z)$ is the total return obtained by using meta learner $f$ on reward function $r_z$, and $f^* \in \arg\max \mathbb{E}_p[R(f^*, r_z)]$ is an optimal meta-learner. This is equivalent to the expected reward objective used by most meta-RL methods (Finn et al., 2017a; Duan et al., 2016b). Note that the behavior of the optimal meta-learner depends on the specific task distribution. For example, the optimal behavior for manipulation tasks involves moving a robot's arms, while the optimal behavior for locomotion tasks involves moving a robot's legs. Optimizing Equation 1 requires inferring new tasks and acquiring the policy for each task. While learning an internal model of the environment might assist in these steps, an optimal meta-learner would also need to know how to leverage this model. In the remainder, we will abstract away in the internal representation of the meta-learner and instead focus on the effect of the training task distribution on the behavior induced in an optimal meta-learner, and how this suggests a method to construct unsupervised meta-learning algorithms.

## 3.3 Special Case: Goal-Reaching Tasks

We will now derive an optimal *unsupervised* meta-learner for the special case of goal reaching tasks, and then generalize this approach to solve arbitrary tasks in Section 3.4. In both, we will restrict our attention to CMPs with deterministic dynamics. In this goal-reaching setting, we consider episodes with finite horizon $T$ and a discount factor of $\gamma = 1$. Tasks correspond to reaching an unknown goal state $s_g$. We will only consider the agent's state at the last time step in each episode, so the (unknown) reward function is always of the form

$$r_g(s_t) \triangleq \mathbb{1}(t = T) \cdot \mathbb{1}(s_t = g).$$

We will first assume that goal states are drawn from some known distribution $p(s_g)$, and later will show how we can remove this assumption. We define $\rho_\pi^T(s)$ as the probability that policy $\pi$ visits

state $s$ at time step $t = T$. If $s_g$ is the true goal, then the event that the policy $\pi$ reaches $s_g$ at the final step of an episode is a Bernoulli random variable with parameter $p = \rho_\pi^T(s_g)$. Thus, the expected *hitting time* of this goal state is

$$\text{HITTINGTIME}_\pi(s_g) = \frac{1}{\rho_\pi^T(s_g)}$$

We now define a meta-learner $f_\pi$ in terms of an exploration policy $\pi$. Before the goal is found, $f_\pi$ uses policy $\pi$ to explore. Once the goal $s_g$ is found, the meta-learner takes actions to always return to state $s_g$. Meta-learner $f_\pi$ incurs one unit of regret for each step before it has found the goal, and zero regret afterwards. The expected cumulative regret is therefore the expectation of the hitting time, taken with respect to $p(s_g) = p(r_g)$:

$$\text{REGRET}(f_\pi, p(r_g)) = \int \text{HITTINGTIME}_\pi(s_g) p(s_g) ds_g = \int \frac{p(s_g)}{\rho_\pi^T(s_g)} ds_g \qquad (2)$$

In this special case, an optimal meta-learner as defined in Section 3.2 will explore for a number of episodes until it finds the goal state. After the meta-learner finds the goal state, it would always choose the trajectory that reaches that goal state under deterministic dynamics. Thus, the cumulative regret of the meta-learner is the number of episodes required to find the goal state. By our assumption that the meta-learner only receives information about the task if it has reached the goal state at the end of the episode, the meta-learner cannot use information about multiple goals within a single episode. We can minimize the regret in Equation 2 w.r.t. the marginal distribution $\rho_\pi^T$. Using the calculus of variations (for more details refer to Appendix C in Lee et al. (2019)), the (exploration) policy for the optimal meta-learner, $\pi^*$, satisfies:

$$\rho_{\pi^*}^T(s_g) = \frac{\sqrt{p(s_g)}}{\int \sqrt{p(s_g')} ds_g'}. \qquad (3)$$

The analysis so far tells us how to obtain the optimal meta-learner if were were given the goal sampling distribution, $p(s_g)$. If we do not know this distribution, then we cannot compute the optimal policy using Equation 3. In this case, we resort to bounding the *worst-case* regret of our policy:

$$\text{REGRET}_{\text{WC}} = \min_\pi \max_{p(s_g)} \text{REGRET}(f_\pi, p(r_g)) \qquad (4)$$

**Lemma 1.** *Let $\pi$ be a policy for which $\rho_\pi^T(s)$ is uniform. Then $f_\pi$ has lowest worst-case regret.*

The proof is in Appendix A. Given this result, we know that the exploration policy of the optimal meta-learner should have a uniform state marginal distribution. While Lemma 1 is only directly applicable in settings where there exists a policy that achieves a uniform state marginal distribution, we might expect the intuition to carry over to other settings. The minimax optimal meta-learner corresponds to exploring over a uniform distribution over goal states, so we can acquire this meta-learner by training on a goal-reaching task distribution where the goals are uniformly distributed. Constructing this distribution is hard, especially in high-dimensional state spaces. *How might we propose uniformly-distributed goals as tasks during unsupervised meta-training?* The key idea is that this uniform goal proposal distribution can be obtained by maximizing the mutual information between $z$ and the final state $s_T$:

$$\max_{\mu(s_T, z) \in \mathcal{P}_\mu} I_\mu(s_T; z) \triangleq \mathcal{H}_\mu[s_T] - \mathcal{H}_\mu[s_T \mid z], \qquad (5)$$

where $\mu$ denotes a probability distribution over terminal states $s_T$ and latent variables $z$, and $\mathcal{P}_\mu$ is a subset of such distributions. Observe that this objective contains two competing terms. The first term, $\mathcal{H}_\mu[s_T]$, is maximized when the marginal distribution $\mu(s_T)$ is uniform. The second term, $\mathcal{H}_\mu[s_T \mid z]$ is maximized when $\mu(s_T \mid z)$ is a Dirac distribution. The distribution $\mu$ that optimizes Equation 5 is one with a uniform marginal distribution over terminal states (proof in Appendix A):

**Lemma 2.** *Assume there exists some distribution $\mu^* \in \mathcal{P}_\mu$ such that $\mu^*(s_T) = \text{UNIF}(\mathcal{S})$ is uniform over goal states and $\mu^*(s_T \mid z) \overset{d}{=} \mathbb{1}(s_T = s_z)$ is a Dirac distribution, centered at a state $s_z$ specified by the latent $z$. Then any distribution $\mu$ that maximizes mutual information $I_\mu(s_T; z)$ satisfies $\mu(s_T) = \text{UNIF}(\mathcal{S})$ and $\mu(s_T \mid z) \overset{d}{=} \mathbb{1}(s_T = s_z)$.*

The key idea is that a joint distribution $\mu(s_T, z)$ can be defined implicitly via a latent-conditioned policy. This policy is *not* a meta-learned model, but rather will become part of the task proposal mechanism. For a given prior $\mu(z)$ and latent-conditioned policy $\mu(a \mid s, z)$, the joint likelihood is $\mu(\tau, z) = \mu(z)p(s_1) \prod_t p(s_{t+1} \mid s_t, a_t)\mu(a_t \mid s_t, z)$, and the marginal likelihood is simply given by $\mu(s_T, z) = \int \mu(\tau, z)ds_1a_1 \cdots a_{T-1}$. We define $\mathcal{P}_\mu$ as all distributions $\mu(s_T, z)$ that can be constructed in such a way using a Markovian policy. Note that the assumption in Lemma 2 does not always hold, as stochasticity in the environment may mean that $\mu(s_T \mid z)$ is never a Dirac distribution, or the existence of unreachable states may mean that $\mu(s_T)$ can never be uniform.

Finally, we construct a distribution over tasks from the joint distribution $\mu$. We then define our task proposal distribution by sampling $z \sim p(z)$ and using the corresponding reward function $r_z(s_T, a_T) \triangleq \log p(s_T \mid z)$. This reward function does not depend on the action $a_T$. In the case when both the prior $\mu(z)$ and the marginal $\mu(s_T)$ are uniform, this reward function is equivalent to the DIAYN reward function $\log \mu(z \mid s_T)$, up to additive constants. Since each latent $z$ corresponds to a single state $s_T$ when the mutual information is maximized, our task distribution corresponds exactly to sampling goals uniformly when $z$ is sampled uniformly. As argued above, successfully meta-learning on the uniform goal distribution produces the minimax-optimal meta-learner.[3]

### 3.4 General Case: Trajectory-Matching Tasks

To extend the analysis in the previous section to the general case, and thereby derive a framework for optimal unsupervised meta-learning, we will consider "trajectory-matching" tasks. These tasks are a trajectory-based generalization of goal reaching: while goal reaching tasks only provide a positive reward when the policy reaches the goal state, trajectory-matching tasks only provide a positive reward when the policy executes the optimal trajectory.The trajectory matching case is more general because, while trajectory matching can represent different goal-reaching tasks, it can also represent tasks that are not simply goal reaching, such as reaching a goal while avoiding a dangerous region or reaching a goal in a particular way. As before, we will restrict our attention to CMPs with deterministic dynamics. These non-Markovian tasks essentially amount to a problem where an RL algorithm must "guess" the optimal policy, and only receives a reward if its behavior is perfectly consistent with that optimal policy. We will show that mutual information between $z$ and *trajectories* yields the minimum regret solution in this case, and then show that unsupervised meta-learning for the trajectory-matching task is at least as hard as unsupervised meta-learning for general tasks (though, in practice, general tasks may be easier).

Formally, we define a distribution of trajectory-matching tasks by a distribution over goal trajectories, $p(\tau^*)$. For each goal trajectory $\tau^*$, the corresponding trajectory-level reward function is

$$r_\tau^*(\tau) \triangleq \mathbb{1}(\tau = \tau^*) \tag{6}$$

Analogous to before, we define the hitting time as the expected number of episodes to match the target trajectory:

$$\text{HITTINGTIME}_\pi(\tau^*) = \frac{1}{\pi(\tau^*)} \tag{7}$$

We then define regret as the expected hitting time:

$$\text{REGRET}(f_\pi, p(r_\tau)) = \int \text{HITTINGTIME}_\pi(\tau)p(\tau)d\tau = \int \frac{p(\tau)}{\pi(\tau)}d\tau \tag{8}$$

Using the same derivation as before, the exploration policy for the optimal meta-learner is

$$\pi^*(\tau) = \frac{\sqrt{p(\tau)}}{\int \sqrt{p(\tau')}d\tau'}. \tag{9}$$

However, obtaining such a policy requires knowing the trajectory distribution $p(\tau)$. In the setting where $p(\tau)$ is unknown, the minimax policy is simply uniform:

---

[3]It is important to note that the task-proposal policy $\mu$ *is not* an optimal meta-learner. For example, in a k-armed bandit, the optimal meta-learner would simply enumerate over all arms and find the best arm in $k/2$ steps, whereas the task-proposal policy would randomly sample arms and require approximately $k \log(k)/2$ steps to find the best arm.

**Lemma 3.** *Let $\pi$ be a policy for which $\pi(\tau)$ is uniform. Then $f_\pi$ has lowest worst-case regret.*

How can we acquire a policy with a uniform trajectory distribution? Repeating the steps above, we learn a collection of skills using a *trajectory-level* mutual information objective:

$$I(\tau; z) = \mathcal{H}[\tau] - \mathcal{H}[\tau \mid z] \tag{10}$$

Using the same reasoning as Section 3.3, the optimal policy for this objective has a uniform distribution over trajectories that, conditioned on a particular latent $z$, deterministically produces a single trajectory in a deterministic CMP. Analogous to Section 3.3, we define a distribution over reward functions as $r_z(\tau) \triangleq \log p(\tau \mid z)$. At optimality, each $z$ corresponds to exactly one trajectory $\tau_z$, so the reward function $r_z(\tau)$ simply indicates whether $\tau$ is equal to $\tau_z$. Since the distribution over trajectories $\int p(\tau \mid z)p(z)dz$ is uniform at the optimum, the distribution of reward functions $r_z$ corresponds to a uniform distribution over trajectories. Thus, meta-learning on the rewards from trajectory-level mutual information results in the minimax-optimal meta-learner.

Now that we have derived the optimal meta-learner for trajectory-matching tasks, observe that trajectory-matching is a super-set of the problem of optimizing any possible Markovian reward function at test-time. For a given initial state distribution, each reward function is optimized by a particular trajectory. However, trajectories produced by a non-Markovian policy (i.e., a policy with memory) are not necessarily the unique optimum for any Markovian reward function. Let $R_\tau$ denote the set of trajectory-level reward functions, and $R_{s,a}$ denote the set of all state-action level reward functions. Bounding the worst-case regret on $R_\tau$ minimizes an upper bound on the worst-case regret on $R_{s,a}$:

$$\min_{r_\tau \in R_\tau} \mathbb{E}_\pi \left[ r_\tau(\tau) \right] \leq \min_{r \in R_{s,a}} \mathbb{E}_\pi \left[ \sum_t r(s_t, a_t) \right] \qquad \text{for all policies } \pi.$$

This inequality holds for all policies $\pi$, including the policy that maximizes the LHS. While we aim to maximize the RHS, we only know how to maximize the LHS, which gives us a lower bound on the RHS. This inequality holds for all policies $\pi$, so it also holds for the policy that maximizes the LHS. In general, this bound is loose because the set of all Markovian reward functions is smaller than the set of all trajectory-level reward functions (i.e., trajectory-matching tasks). However, this bound becomes tight when considering meta-learning on the set of all possible (non-Markovian) reward functions.

In the discussion of meta-learning thus far, we have considered tasks where the reward is provided at the last time step $T$ of each episode. In this particular case, the best that an optimal meta-learner can do is go directly to a goal or execute a particular trajectory at every episode according to the optimal exploration policy: $\rho_{\pi^*}^T(s_g) = \frac{\sqrt{p(s_g)}}{\int \sqrt{p(s_g')}ds_g'}$ for goal reaching or $\pi^*(\tau) = \frac{\sqrt{p(\tau)}}{\int \sqrt{p(\tau')}d\tau'}$ for trajectory matching. All intermediate states in the trajectory are uninformative, thus making instances of meta-learning algorithms which explore via schemes like posterior sampling optimal for this class of problems. In the more general case with arbitrary reward functions, intermediate rewards along a trajectory may be informative, and the optimal exploration strategy may be different from posterior sampling (Rothfuss et al., 2019; Duan et al., 2016b; Wang et al., 2016). Nonetheless, the analysis presented in this section provides us insight into the behavior of optimal meta-learning algorithms and allows us to understand the qualities desirable for unsupervised task proposals. The general proposed scheme for unsupervised meta-learning has a number of benefits over standard universal value function and goal reaching style algorithms: (1) it can be applied to arbitrary reward functions going beyond simple goal reaching and (2) it can acquire a significantly more effective exploration strategy than posterior sampling by using intermediate states in a trajectory for intelligent information gathering.

### 3.5 SUMMARY OF ANALYSIS

Through our analysis, we introduced the notion of optimal meta-learners and analyze their exploration behavior and regret on a class of goal reaching problems. We showed that on these problems, when the test-time task distribution is unknown, the optimal meta-training task distribution for minimizing worst-case test-time regret is *uniform* over the space of goals. We also showed that this optimal task distribution can be acquired by a simple mutual information maximization scheme. We subsequently extend the analysis to the more general case of matching arbitrary trajectories, as a proxy for the

more general class of arbitrary reward functions. In the following section, we will discuss how we can derive a practical algorithm for unsupervised meta-learning from this analysis.

## 3.6 A PRACTICAL ALGORITHM

Following the derivation in the previous section, we can instantiate a practical unsupervised meta-RL algorithm by constructing a task proposal mechanism based on a mutual information objective. A variety of different mutual information objectives can be formulated, including mutual information between single states and $z$ (Eysenbach et al., 2018), pairs of start and end states and $z$ (Gregor et al., 2016), and entire trajectories and $z$ (Achiam et al., 2018). We will use DIAYN and leave a full examination of possible mutual information objectives for future work.

DIAYN optimizes mutual information by training a *discriminator* network $D_\phi(z|\cdot)$ that predicts which $z$ was used to generate the states in a given rollout according to a *latent-conditioned policy* $\pi(a|s,z)$. Our task proposal distribution is thus defined by $r_z(s,a) = \log(D_\phi(z|s))$. The complete unsupervised meta-learning algorithm follows the recipe in Figure 1: first, we acquire $r_z(s,a)$ by running DIAYN, which learns $D_\phi(z|s)$ and a latent-conditioned policy $\pi(a|s,z)$ (which is discarded). Then, we use $z \sim p(z)$ to propose tasks $r_z(s,a)$ to a standard

---

**Algorithm 1:** Unsupervised Meta-Reinforcement Learning Pseudocode

**Data:** $\mathcal{M} \setminus R$, an MDP without a reward function
**Result:** a learning algorithm $f : D_\phi \rightarrow \pi$
$D_\phi \leftarrow$ DIAYN() or $D_\phi \leftarrow random$
**while** *not converged* **do**
 Sample latent task variables $z \sim p(z)$
 Extract corresponding task reward functions $r_z(s)$
  using $D_\phi(z|s)$
 Update $f$ using MAML with reward $r_z(s)$

---

meta-RL algorithm. This meta-RL algorithm uses the proposed tasks to learn how to learn, acquiring a fast learn algorithm $f$ which can then learn new tasks quickly. While, in principle, any meta-RL algorithm could be used, the discussion at the end of Section 3.4 suggests that meta-RL methods based on posterior sampling might be suboptimal for some tasks. We use MAML (Finn et al., 2017a) as our meta-learning algorithm. Note that the learning algorithm $f$ returned by MAML is defined simply as running gradient descent using the parameters found by MAML as initialization. While it is not immediately apparent how a parameter initialization can define an entire learning algorithm, insightful prior work Finn & Levine (2017) provides an in-depth discussion of this property. The complete method is summarized in Algorithm 1.

In addition to mutual information maximizing task proposals, we will also consider random task proposals, where we also use a discriminator as the reward, according to $r(s,z) = \log D_{\phi_{rand}}(z|s)$, but where the parameters $\phi_{rand}$ are chosen randomly (i.e., a random weight initialization for a neural network). While such random reward functions are not optimal, we find that they can be used to acquire useful task distributions for simple tasks, though they are not as effective as the tasks become more complicated. This empirically reinforces the claim that unsupervised meta-RL does not in fact violate any "no free lunch" principle – even simple task proposals that cause the meta-learner to explore the CMP can already accelerate learning of new tasks.

## 4 EXPERIMENTAL EVALUATION

In our experiments, we aim to understand whether unsupervised meta-learning as described in Section 3.1 can provide us with an accelerated RL procedure on new tasks. Whereas standard meta-learning requires a hand-specified task distribution at meta-training time, unsupervised meta-learning learns the task distribution

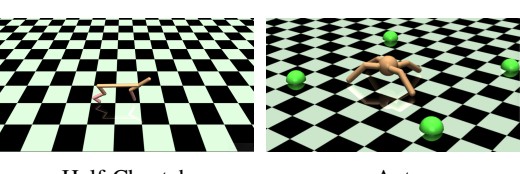

Half-Cheetah          Ant

through unsupervised interaction with the environment. A fair baseline that likewise uses requires *no reward supervision* at training time, and only uses rewards at test time, is learning via RL from scratch without any meta-learning. As an upper bound, we include the *unfair* comparison to a standard meta-learning approach, where the meta-training distribution is manually designed. This method has access to a hand-specified task distribution that is not available to our method. We evaluate two variants of our approach: (a) task acquisition based on DIAYN followed by meta-learning using

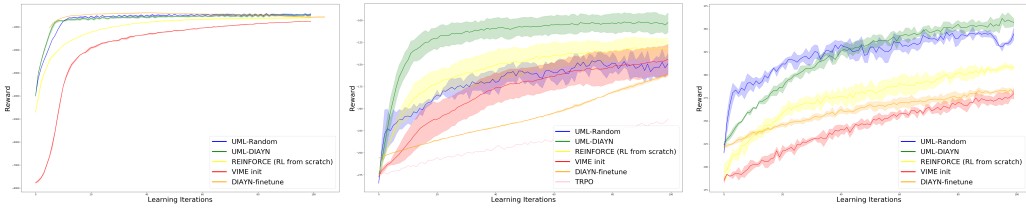

| 2D navigation | Half-Cheetah | Ant |

Figure 3: **Unsupervised meta-learning accelerates learning**: After unsupervised meta-learning, our approach (UML-DIAYN and UML-RANDOM) quickly learns a new task significantly faster than learning from scratch, especially on complex tasks. Learning the task distribution with DIAYN helps more for complex tasks. Results are averaged across 20 evaluation tasks, and 3 random seeds for testing. UML-DIAYN and random also significantly outperform learning with DIAYN initialization or an initialization with a policy pretrained with VIME.

MAML, and (b) task acquisition using a randomly initialized discriminator followed by meta-learning using MAML.

## 4.1 Tasks and Implementation Details

Our experiments study three simulated environments of varying difficulty: 2D point navigation, 2D locomotion using the "HalfCheetah," and 3D locomotion using the "Ant," with the latter two environments are modifications of popular RL benchmarks (Duan et al., 2016a). While the 2D navigation environment allows for direct control of position, HalfCheetah and Ant can only control their center of mass via feedback control with high dimensional actions (6D for HalfCheetah, 8D for Ant) and observations (17D for HalfCheetah, 111D for Ant).

The evaluation tasks, shown in Figure 6, are similar to prior work (Finn et al., 2017a; Pong et al., 2018): 2D navigation and ant require navigating to goal positions, while the half cheetah must run at different goal velocities. These tasks are not accessible to our algorithm during meta-training. We used the default hyperparameters for MAML across all tasks, varying the meta-batch size according to the number of skills: 50 for pointmass, 20 for cheetah, and 20 ant. We found that the default architecture, a 2 layer MLP with 300 units each and ReLU non-linearities, worked quite well for meta-training. We also used the default hyperparameters for DIAYN to acquire skills. We swept over learning rates for learning from scratch via vanilla policy gradient, and found that using ADAM with adaptive step size is the most stable and quick at learning.

## 4.2 Fast Adaptation after Unsupervised Meta Learning

The comparison between the two variants of unsupervised meta-learning and learning from scratch is shown in Figure 3. We also add a comparison to VIME (Houthooft et al., 2016), a standard novelty-based exploration method, where we pretrain a policy with the VIME reward and then finetune it on the meta-test tasks. In all cases, the UML-DIAYN variant of unsupervised meta-learning produces an RL procedure that outperforms RL from scratch and VIME-init, suggesting that unsupervised interaction with the environment and meta-learning is effective in producing environment-specific but task-agnostic priors that accelerate learning on new, previously unseen tasks. The comparison with VIME shows that the speed of learning is not just about exploration but is indeed about fast adaptation. In our experiments thus far, UML-DIAYN always performs better than learning from scratch, although the benefit varies across tasks depending on the actual performance of DIAYN. We added additional comparisons to using TRPO as a stronger optimizer from scratch with the same trends being observed. We also compared with a baseline of simply initializing from a DIAYN trained contextual policy, and then finetuning the best skill with the actual task reward. From the comparisons in Fig 3, it is apparent that this works poorly as compared with UMRL variants.

Interestingly, in many cases (in Figure 4) the performance of unsupervised meta-learning with DIAYN matches that of the hand-designed task distribution. We see that on the 2D navigation task, while handcrafted meta-learning is able to learn very quickly initially, it performs similarly after 100 steps. For the cheetah environment as well, handcrafted meta-learning is able to learn very quickly to start off, but is quickly matched by unsupervised meta-RL with DIAYN. On the ant task, we see that hand-crafted meta-learning does do better than UML-DIAYN, likely because the task distribution is

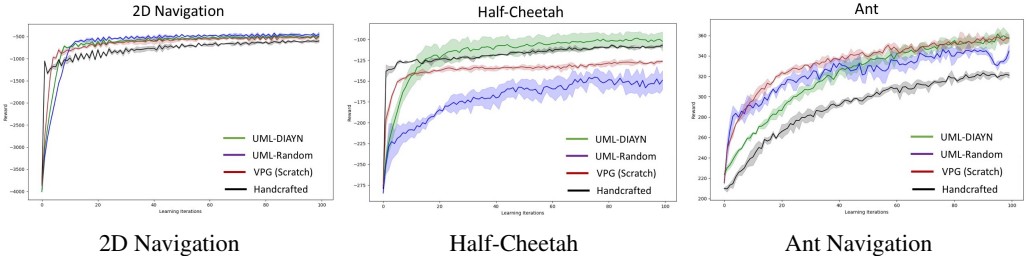

| 2D Navigation | Half-Cheetah | Ant Navigation |
|---|---|---|

Figure 4: **Comparison with handcrafted tasks**: Unsupervised meta-learning (UML-DIAYN) is competitive with meta-training on handcrafted reward functions (i.e., an oracle). A misspecified, handcrafted meta-training task distribution often performs worse, illustrating the benefits of learning the task distribution.

more challenging, and a better unsupervised task proposal algorithm would improve the performance of a meta-learner.

The comparison between the two unsupervised meta-learning variants is also illuminating: while the DIAYN-based variant of our method generally achieves the best performance, even the random discriminator is often able to provide a sufficient diversity of tasks to produce meaningful acceleration over learning from scratch in the case of 2D navigation and ant. This result has two interesting implications. First, it suggests that unsupervised meta-learning is an effective tool for learning an environment prior. Although the performance of unsupervised meta-learning can be improved with better coverage using DIAYN (as seen in Figure 3), even the random discriminator version provides competitive advantages over learning from scratch. Second, the comparison provides a clue for identifying the source of the structure learned through unsupervised meta-learning: though the particular task distribution has an effect on performance, simply interacting with the environment (without structured objectives, using a random discriminator) already allows meta-RL to learn effective adaptation strategies in a given environment. That is, the performance cannot be explained only by the unsupervised procedure (DIAYN) capturing the right task distribution. We also provide an analysis of the task distributions acquired by the DIAYN procedure in Appendix B.1.

## 5 DISCUSSION AND FUTURE WORK

We presented an unsupervised approach to meta-RL, where meta-learning is used to acquire an efficient RL procedure without requiring hand-specified task distributions for meta-training. This approach accelerates RL without relying on the manual supervision required for conventional meta-learning algorithms. We provide a theoretical derivation that argues that task proposals based on mutual information maximization can provide for a minimum worst-case regret meta-learner, under certain assumptions. We then instantiate an approximation to the theoretically-motivated method by building on recently developed unsupervised task proposal and meta-learning algorithms. Our experiments indicate that unsupervised meta-RL can accelerate learning on a range of tasks, outperforming learning from scratch and often matching the performance of meta-learning from hand-specified task distributions.

As our work is the first foray into unsupervised meta-RL, our approach opens a number of questions about unsupervised meta-learning algorithms. One limitation of our analysis is that it only considers deterministic dynamics, and only considers task distributions where posterior sampling is optimal. Extending our analysis to stochastic dynamics and more realistic task distributions may allow unsupervised meta-RL to acquire learning algorithms that can explore and adapt more intelligently, and more effectively solve real-world tasks.

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

## A   PROOFS

**Lemma 1** Let $\pi$ be a policy for which $\rho_\pi^T(s)$ is uniform. Then $\pi$ has lowest worst-case regret.

*Proof of Lemma 1.* To begin, we note that all goal distributions $p(s_g)$ have equal regret for policies where $\rho_\pi^T(s) = 1/|\mathcal{S}|$ is uniform:

$$\text{REGRET}_p(\pi) = \int \frac{p(s_g)}{\rho_\pi^T(s_g)} ds_g = \int \frac{p(s_g)}{1/|\mathcal{S}|} ds_g = |\mathcal{S}|$$

Now, consider a policy $\pi'$ for which $\rho_\pi^T(s)$ is not uniform. For simplicity, we will assume that the argmin is unique, though the proof holds for non-unique argmins as well. The worst-case goal distribution will choose the state $s^-$ where that the policy is least likely to visit:

$$p^-(s_g) \triangleq \mathbb{1}(s_g = \arg\min_s \rho_\pi^T(s))$$

Thus, the worst-case regret for policy $\pi'$ is strictly greater than the regret for a uniform $\pi$:

$$\max_p \text{REGRET}_p(\pi) = \text{REGRET}_{p^-}(\pi) = \int \frac{\mathbb{1}(s_g = \arg\min_s \rho_\pi^T(s))}{\rho_\pi^T(s_g)} ds_g = \frac{1}{\min_s \rho_{\pi'}^T(s)} > |\mathcal{S}|$$

(11)

Thus, a policy $\pi'$ for which $\rho_\pi^T$ is non-uniform cannot be minimax, so the optimal policy has a uniform marginal $\rho_\pi^T$. $\qquad\square$

**Lemma 2**: Mutual information $I(s_T; z)$ is maximized by a task distribution $p(s_g)$ which is uniform over goal states.

*Proof of Lemma 2.* We define a latent variable model, where we sample a latent variable $z$ from a uniform prior $p(z)$ and sample goals from a conditional distribution $p(s_T \mid z)$. To begin, note that the mutual information can be written as a difference of entropies:

$$I_p(s_T; z) = \mathcal{H}_p[s_T] - \mathcal{H}_p[s_T \mid z]$$

The conditional entropy $\mathcal{H}_p[s_T \mid z]$ attains the smallest possible value (zero) when each latent variable $z$ corresponds to exactly one final state, $s_z$. In contrast, the marginal entropy $\mathcal{H}_p[s_T]$ attains the largest possible value ($\log|\mathcal{S}|$) when the marginal distribution $p(s_T) = \int p(s_T \mid z)p(z)dz$ is uniform. Thus, a task uniform distribution $p(s_g)$ maximizes $I(s_T; z)$. Note that for any non-uniform task distribution $q(s_T)$, we have $\mathcal{H}_q[s_T] < \mathcal{H}_p[s_T]$. Since the conditional entropy $\mathcal{H}_p[s_T \mid z]$ is zero, no distribution can achieve a smaller conditional entropy. This, for all non-uniform task distributions $q$, we have $I_q(s_T; z) < I_p(s_T; z)$. Thus, the optimal task distribution must be uniform. $\qquad\square$

## B   ABLATIONS

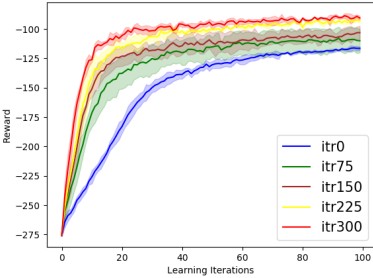

Figure 5: Analysis of effect of additional meta-training on meta-test time learning of new tasks. For larger iterations of meta-trained policies, we have improved test time performance, showing that additional meta-training is beneficial.

To understand the method performance more clearly, we also add an ablation study where we compare the meta-test performance of policies at different iterations along meta-training. This shows the effect

that additional meta-training has on the fast learning performance for new tasks. This comparison is shown in Figure 5. As can be seen here, at iteration 0 of meta-training the policy is not a very good initialization for learning new tasks. As we move further along the meta-training process, we see that the meta-learned initialization becomes more and more effective at learning new tasks. This shows a clear correlation between additional meta-training and improved meta test-time performance.

## B.1 Analysis of Learned Task Distributions

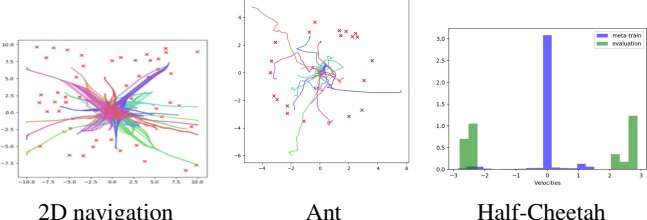

|  2D navigation  |  Ant  |  Half-Cheetah  |

Figure 6: **Learned meta-training task distribution and evaluation tasks**: We plot the center of mass for various skills discovered by point mass and ant using DIAYN, and a blue histogram of goal velocities for cheetah. Evaluation tasks, which are not provided to the algorithm during meta-training, are plotted as red 'x' for ant and pointmass, and as a green histogram for cheetah. While the meta-training distribution is broad, it does not fully cover the evaluation tasks. Nonetheless, meta-learning on this *learned* task distribution enables efficient learning on a test task distribution.

We can analyze the tasks discovered through unsupervised exploration and compare them to tasks we evaluate on at meta-test time. Figure 6 illustrates these distributions using scatter plots for 2D navigation and the Ant, and a histogram for the HalfCheetah. Note that we visualize dimensions of the state that are relevant for the evaluation tasks – positions and velocities – but these dimensions are *not* specified in any way during unsupervised task acquisition, which operates on the entire state space. Although the tasks proposed via unsupervised exploration provide fairly broad coverage, they are clearly quite distinct from the meta-test tasks, suggesting the approach can tolerate considerable distributional shift. Qualitatively, many of the tasks proposed via unsupervised exploration such as jumping and falling that are not relevant for the evaluation tasks. Our choice of the evaluation tasks was largely based on prior work, and therefore not tailored to this exploration procedure. The results for unsupervised meta-RL therefore suggest quite strongly that unsupervised task acquisition can provide an effective meta-training set, at least for MAML, even when evaluating on tasks that do not closely match the discovered task distribution.

## C Hyperparameter Details

For all our experiments, we used DIAYN to acquire the task proposals using 20 skills for half-cheetah and for ant and 50 skills for the 2D navigation. We ran the domains using the standard DIAYN hyperparameters described in `https://github.com/ben-eysenbach/sac` to acquire task proposals. These proposals were then fed into the MAML algorithm `https://github.com/cbfinn/maml_rl`, with inner learning rate 0.1, meta learning rate 0.01, inner batch size 40, outer batch size 20, path length 100, using 2 layer networks with 300 units each with ReLu nonlinearities. The test time learning is done with the same parameters for the UMRL variants, and done using REINFORCE with the Adam optimizer for the comparison with learning from scratch.

