# OpenReview forum: "Unsupervised Meta-Learning for Reinforcement Learning"
_ICLR.cc/2020/Conference — Reject_

### Official Review · AnonReviewer2 · 2019-10-23
**Official Blind Review #2**

**Rating:** 3

**Review:**

The paper develops a meta-learning approach for improving sample efficiency of learning different tasks in the same environment. The author formulates the meta goal as minimizing the expected regret under the worst case, which happens when all the tasks are uniformly distributed. The paper introduces two types of tasks: goal-reaching task and a more general trajectory matching task. Then the author introduces a meta-learning algorithm to minimize the regret by learning the reward function under different sampled tasks. The paper is interesting. Below are my questions/concerns.

1. Why trajectory matching is considered as more general? Intuitively, trajectory matching is more restricted in that whenever an agent can match the optimal trajectory, it should also reach the goal state.

2. The theoretical results (lemma 2, 3) actually indicates that the previous work universal value function approximator can optimize the proposed meta learning objective with theoretical convergence guarantee in tabular case by learning the value function Q(s, g, a) where s is a state, g is goal state, a is an action (as long as s and g are visited infinitely often) . As a result, why is it necessary to introduce meta-learning approach? Why not simply learn universal value functions?

3. The experimental results are not very persuasive. What is the VPG algorithm used? And if you run the algorithm longer, is it finally worse than learning from scratch? Option learning methods/universal value function can be added as baselines.

**Experience Assessment:**

I have read many papers in this area.

**Review Assessment: Checking Correctness Of Derivations And Theory:**

I did not assess the derivations or theory.

**Review Assessment: Checking Correctness Of Experiments:**

I assessed the sensibility of the experiments.

**Review Assessment: Thoroughness In Paper Reading:**

I read the paper at least twice and used my best judgement in assessing the paper.

---

> ### Author Response · Authors · 2019-11-12
> **Response to R2**
>
> We thank the reviewer for their feedback and suggestions! We have added clarifications to the paper based on the suggestions and questions (refer to Section 3.4), as well as added additional comparisons (Section 4.2, Fig 3). Please find detailed comments below:
>
> “Why trajectory matching is considered as more general? “
> -> While it is true that whenever a policy matches a trajectory, it reaches the goal state, the trajectory matching case is more general because, while trajectory matching can represent different goal-reaching tasks, it can also represent tasks that are not simply goal reaching, such as reaching a goal while avoiding a dangerous region or reaching a goal in a particular way. We have added this discussion to Section 3.4
>
> “why is it necessary to introduce meta-learning approach? Why not simply learn universal value functions?”
> ->This is a very interesting question, however it is not specific to the paradigm of unsupervised meta-learning, the same can be asked about any meta-learning algorithm. Using a meta-learner has two major advantages: first, it allows us to operate in the cases where the exact g is not specified, but the task is simply specified through the reward, which is natural in many scenarios. In these cases, a meta-learner would also acquire a more optimal exploration strategy than simply trying different g like a learned Q(s,a,g) would need to do. Second, the meta-learner can optimize an arbitrary reward function and not simply the goal reaching reward which Q(s,a,g) would be restricted to. A similar discussion is added to Section 3.4.
>
> “The experimental results are not very persuasive. What is the VPG algorithm used?”
> -> The experimental comparisons are with other algorithms that learn from scratch to ensure a fair comparison, since the amount of reward supervision is the same as our algorithm. We chose VPG (also called REINFORCE (Williams 92)) to ensure a fair comparison because our meta-learner uses REINFORCE to learn. The choice of specific RL algorithm in the inner loop is orthogonal to the benefits of UMRL, we could replace the inner loop with a more powerful RL algorithm to get similar benefits. We have also included a comparison with another RL algorithm (TRPO) for learning from scratch, which also performs worse than UMRL in Fig 3. We have also added an additional comparison in Fig 3 with finetuning purely from a DIAYN initialization, without any meta-learning involved.

---

### Official Review · AnonReviewer3 · 2019-10-24
**Official Blind Review #3**

**Rating:** 1

**Review:**

Summary: this paper claims to design an unsupervised meta-learning algorithm that does automatically design a task distribution for the target task. The conceptual idea is to propose a task based on mutual information and to train the optimal meta-learner. They also use experiments to show the effectiveness of the proposed approach.

Overall Comments:

I would think this paper requires a major revision. It is written in a very confusing way. Many terms are directly used without a definition. The problem is also not clearly defined. I have tried to understand everything, but I have to give up in Section 3. Overall, I do not think this paper is ready for publication.

Detailed comments:
	• It would benefit a lot if you can clearly define the original meta-learning procedure and then compare that with the one proposed in this paper.
	• Define ”hand-specified” distribution. This word does not make sense if you claim this is the difference between the meta-learning procedure proposed in this paper and the original meta-learning algorithm. In this paper, you used p(z) to specify a task. I would think p(z) is also “hand-specified”.
	• I am not very sure by what you mean for “task-proposal procedure”, “goal-proposal procedure”
	• In the first paragraph of the intro: what do you mean by “specifying a task distribution is tedious”, is specifying p(z) also “tedious”
	• 2nd paragraph of intro: “automate the meta-training process by removing the need for hand-designed meta-training tasks”. Again, why p(z) is not “hand-designed”
	• Why compare with the original meta-RL algorithm on p(z) is not fair?
	• What do you mean by “acquire reinforcement learning procedures”?
	• “Environment”, “task” are not clear when they first appear
	• The word “learn” is used everywhere, and is confusing. E.g. what do you mean by “learn new tasks”, “learn a learning algorithm f”, “learn an optimal policy”, “learn a task distribution” …
	• “Reward functions induced by p(z) and r_z(s,a)”: isn’t r_z(s,a) already a reward function? What is “induced”?
	• What is “meta-training” time?
	• What is “no free lunch theorem”?
	• The “controlled-MDP” setting is actually much easier: perhaps you just need to learn the probability distribution. Then for every r_z, we just solve it. Why not compare with this simple algorithm?
	• “Regret” is not defined when it first appears
	• “The task distribution is defined by a latent variable z and a reward function r_z”: why “distribution” is defined by an r.v.?
	• In (2), “regret” should be the (cost of the algorithm) - (the total cost of an optimal policy) — it is not hitting time
	• (3) is confusing, no derivation is given
	• Based on the usual definition of “regret”, how can a “policy” have low regret? Any fixed “policy” would have linear regret …

**Experience Assessment:**

I have published in this field for several years.

**Review Assessment: Checking Correctness Of Derivations And Theory:**

I assessed the sensibility of the derivations and theory.

**Review Assessment: Checking Correctness Of Experiments:**

I assessed the sensibility of the experiments.

**Review Assessment: Thoroughness In Paper Reading:**

I read the paper at least twice and used my best judgement in assessing the paper.

---

> ### Author Response · Authors · 2019-11-12
> **Response to R3**
>
> We thank the reviewer for their comments and feedback! We would like to clarify the aim of the proposed method: given a particular environment, automatically learn a RL algorithm that can quickly solve tasks in this environment. We have added an explicit definition of the problem statement in Section 3 (paragraph 1) to clarify. Standard meta-learning does not immediately solve this problem, as meta-learning requires a hand-designed task distribution. Our key observation is that, rather than using a hand-designed task distribution, we can automatically acquire this task distribution using an unsupervised skill discovery algorithm (DIAYN). Theoretically, we prove that this method maximizes worst-case regret on new tasks provided at test time.
>
> “ It would benefit a lot if you can clearly define the original meta-learning procedure and then compare that with the one proposed in this paper.”
> -> We have attempted to clarify this in Section 3 and Section 3.2. The key difference is the lack of a known task distribution in our case as opposed to a known task distribution in the standard meta-learning case.
>
> “Define ”hand-specified” distribution”
> -> This means that the reward functions for tasks and the actual distribution of tasks themselves are specified before hand by the human operator and the training and test sets are both drawn from this distribution. We have included this discussion in Section 1 of the paper.
>
> “I am not very sure by what you mean for “task-proposal procedure”, “goal-proposal procedure”
> -> In order to do meta-learning, you’d need a task distribution to sample from. If the task distribution is not hand-specified as in our case, it needs to be proposed by the agent itself. So the procedure (in our case a mutual information style procedure like DIAYN) to generate tasks without supervision, which can then be used for meta-learning is the task-proposal procedure. The goal proposal procedure is a special case of the task-proposal procedure for the case of goal-reaching style tasks. We have included this discussion in Section 3.1 of the updated paper.
>
>
> “In the first paragraph of the intro: what do you mean by “specifying a task distribution is tedious”, is specifying p(z) also “tedious”
> -> In standard meta-learning, the task distribution is specified by manually crafting a large number of tasks. This involves manually writing down reward functions. We believe this is tedious and time-consuming. In automated skill discovery mechanisms, the prior p(z) is typically uniform, and is therefore trivial to define, as now discussed in Section 3.1.
>
> “2nd paragraph of intro: “automate the meta-training process by removing the need for hand-designed meta-training tasks”. Again, why p(z) is not “hand-designed”
> - > In most unsupervised skill discovery algorithms, including the DIAYN algorithm used in our method, p(z) is simply uniform. Hence, while it is chosen manually, it is trivial to "design," analogously to the prior in a latent variable model.
>
> “What do you mean by “acquire reinforcement learning procedures”?”
> -> This is following the paradigm of meta-reinforcement learning. A meta-reinforcement learning algorithm learns how to learn: it uses a set of meta-training tasks to learn a learning function f, which can then learn a new task. We refer to this learned learning function f as an "acquired reinforcement learning procedure," following prior work, such as MAML (Finn et al) and RL2 (Duan et al). We have included this in Section 3.1.
>
> “ Why compare with the original meta-RL algorithm on p(z) is not fair? “
> -> p(z) does not define a task distribution by itself, in the same way that the prior in a latent variable model does not by itself define a likelihood. A task is given by a reward function, i.e. r(s, a, z), while p(z) is just a uniform prior on a latent variable. It needs to have rewards defined in order to meta-RL
>
> “The controlled-MDP” setting is actually much easier”:
> -> In the absence of a reward function, it is not clear what rewards to be optimizing your policy on. Only once a CMP is combined with a reward do we get a MDP which can be solved with an RL problem.
>
> We have added definitions for the terms requested to the paper and also below:
> “meta-training time”: process of learning the fast RL algorithm via a meta-RL algorithm such as MAML or RL2. Added to Section 3.1
>
> “No-free lunch theorem”: This states “All algorithms that search for an extremum of a cost function perform exactly the same when averaged over all possible cost functions.” Please refer to Wolpert et al for more detail.
>
> “Regret”: As described by the optimal meta-learner in Section 3.2 (Equation 1), the regret of the policy is indeed the hitting time, because once it finds the right goal/trajectory, the optimal meta-learner would simply keep going to the goal or replicating the trajectory. By saying that a policy has low regret, we mean that the (learned) learning algorithm f has low regret. We have corrected this in the text.

---

### Official Review · AnonReviewer1 · 2019-10-30
**Official Blind Review #1**

**Rating:** 3

**Review:**

# Summary of the paper:

This paper formulates conceptually the unsupervised meta-RL problem (to learn a policy without access to any reward function) as a minimization of the expected regret over tasks, and instantiate an algorithm based on DIAYN (Eysenbach et al., 2018) and MAML Finn et al. (2017a).

# Brief explanation of my rating:

1. *Novelty*: Mutual information based unsupervised RL was proposed by DIAYN (Eysenbach et al., 2018). Meta-model was also considered by DIAYN (Eysenbach et al., 2018), in which they call it "skill".
2. *Technical contributions*: Sec 3.1-3.4 try to justify DIAYN. However, the reasoning is not sufficiently rigorous and the proposed Algorithm 1 is inconsistent with the theory built up in these sections.
3. The *writing* can be improved a lot -- it's not easy to guess what the author was trying to say until I read DIAYN (Eysenbach et al., 2018).
4. The key ingredient is missing -- the learning procedure f, which was mentioned in eq.(1) and Algorithm 1, but the details are never specified. It is impossible to reproduce the algorithm based on the description in the paper.
4. The same *experiments* are conducted in DIAYN (Eysenbach et al., 2018). I am still confused on why we suddenly should use meta-RL.

# Comments:

1. Why we should consider regret? What is the relation between (1) & (4)? It's quite strange you start with (1) but turn to something else, i.e., (4), quickly.
2. "This policy induces a distribution over terminal states, p(s_T | z)" Why?
3. What are you optimizing over in (5)?  The statement in Lemma 2 says "I(s_T; z) maximized by a task distribution p(s_g)". However, you are only able to control p(s_T | z), not the marginal distribution p(s_T). The statement of Lemma should be made more clear.
4. The definition of the reward function: r_z(s_T, a_T) = log p(S_T | z), which is independent of the action a_T?
5. In Algorithm 1, the reward reuse the definition of DIAYN -- log D(z | s),  but which is different from log p(S_T | z). Could you elaborate this?
6. What is the definition of Markovian reward? Why does the inequality on page 6 hold?

**Experience Assessment:**

I have published one or two papers in this area.

**Review Assessment: Checking Correctness Of Derivations And Theory:**

I carefully checked the derivations and theory.

**Review Assessment: Checking Correctness Of Experiments:**

I assessed the sensibility of the experiments.

**Review Assessment: Thoroughness In Paper Reading:**

I read the paper at least twice and used my best judgement in assessing the paper.

---

> ### Author Response · Authors · 2019-11-12
> **Response to R1 (1/2)**
>
> We thank the reviewer for their feedback and suggestions! Below, we emphasize that the setting we consider is actually quite different from that considered in DIAYN. We have updated the paper to clarify the questions raised above, including a new empirical comparison to DIAYN. Please let us know if this addresses your concerns, or if there are further issues you would like us to attempt to fix.
>
> “*Novelty*”
> -> We are not proposing a new unsupervised skill discovery scheme or a new meta-learning algorithm. Rather, we argue that previously proposed MI-based skill discovery schemes (e.g., DIAYN) can both practically and theoretically allow us to apply meta-learning to tasks without manually specifying a task distribution. To clarify the contribution of the work, we have added a new comparison with DIAYN in Fig 3. The scheme in DIAYN does not learn a fast learning algorithm that solves new tasks from reward signals, it simply provides a good set of skills that cover the state space — how to select which of these skills to then use to solve a new task is a separate problem. The actual reinforcement learning procedure to learn a new task from this initialization can still be very slow (see Fig 3). In contrast, meta-learning algorithms can learn to learn new tasks very quickly, but require manually provided task distributions to meta-train on. We argue that using MI-based skill discovery methods like DIAYN, together with meta-learning, addresses the shortcomings of both methods, allowing for fast adaptation without requiring manually provided task distributions. We believe that this observation is novel and relevant.
>
> “*Technical contributions*”
> -> Sections 3.1 - 3.4 aim to justify why DIAYN — or any other MI-based skill discover method -- is a reasonable choice for an unsupervised meta-learning task proposal mechanism. It doesn’t try to justify why DIAYN works, but merely why using that objective provides a meta-learner which has the lowest worst-case regret. We would emphasize that Algorithm 1 does in fact implement an approximation to the principled procedure outlined in Section 3.4, with the following approximations: DIAYN considers states along a trajectory to be conditionally independent and treats them as a bag of states for discrimination, rather than discriminating on entire trajectories. As always, there are a number of approximations that are needed to actually instantiate the theoretically principled method, but we do not believe that the approximations we employ in this regard are especially egregious and this has also been discussed in prior work (Variational Option Discovery Algorithms, Achiam et al 2018). However, if there are specific inconsistencies that you believe would cause major issues, we would be happy to discuss this!
>
> “The *writing* can be improved a lot”
> -> We have rewritten much of the analysis (Section 3), as well as sentences throughout the rest of the paper, to clarify the writing. If there are specific points that would benefit from further clarification, please let us know! We would be happy to make whatever modifications further clarify the exposition.
>
> “The key ingredient is missing -- the learning procedure f, which was mentioned in eq.(1) and Algorithm 1, but the details are never specified. It is impossible to reproduce the algorithm based on the description in the paper. “
> -> We have clarified Section 3.6 to describe the resulting learning procedure. The learning procedure that is returned by MAML is defined by running gradient descent, starting with the initial parameters found by MAML (See “Meta-Learning and Universality: Deep Representations and Gradient Descent can Approximate any Learning Algorithm” (Finn & Levine) for more discussion). In short, the proposed algorithm uses DIAYN to generate a set of self-proposed tasks in an environment, uses the discriminator from DIAYN to provide a reward function to MAML, and then returns a learning procedure f defined by running gradient descent starting initialized at the weights found by MAML. We have also added additional experimental details to Appendix C.
>
> “The same *experiments* are conducted in DIAYN (Eysenbach et al., 2018).”
> -> The experiments in this work are different from those conducted in DIAYN. The plots in the main paper (Fig 3 and Fig 4) consider a meta-learning setting, a setting not considered in DIAYN. While DIAYN does indeed learn a good set of initial skills, subsequent reinforcement learning can still be quite slow. We have added a comparison (Fig 3) to simply initializing with DIAYN and running finetuning as described in Eysenbach et al, and we find that this performs quite poorly on our test-time tasks.

---

> > ### Author Response · Authors · 2019-11-12
> > **Response to R1 (2/2)**
> >
> > Response to Comments:
> >
> > 1. Regret is simply a standard metric for measuring learning speed. Equation 1 considers meta-learning in settings where the task distribution is known. The key contribution of our paper is to consider the setting where the task distribution is not known. Equation 4 introduces the metric we consider in this setting: regret under the worst-case task distribution.
> > 2. This is simply a definition for our didactic goal-reaching case. We have clarified the wording in Section 3.3 to indicate that this is a definition.
> > 3. We maximize the mutual information w.r.t. the joint distribution over latent z and terminal state s_T. We have rewritten Section 3.3, including Lemma 1, to clarify this point.
> > 4. Yes. We have added a sentence after this definition to clarify.
> > 5. In the case where the prior p(z) is uniform and the marginal p(s_T) is uniform (i.e., when the mutual information is maximized), the two reward functions are equivalent, up to an additive constant: log p(z | s) = log p(s | z) + log p(z) - log p(s)
> > 6. A Markovian reward function is one that depends only on the current state and action. Reward functions that depend on (say), that action you took 5 steps prior are not Markovian. Not all reward functions are Markovian. The inequality on page 6 say: a policy that does well on all reward functions is guaranteed to do at least as well on the subset of reward functions which are Markovian.

---

### Decision · Program_Chairs · 2019-12-19

**Decision:**

Reject

**Comment:**

The paper discusses the relevant topic of unsupervised meta-learning in an RL setting. The topic is an interesting one, but the writing and motivation could be much clearer. I advise the authors to make a few more iterations on the paper taking into account the reviewers' comments and then resubmit to a different venue.